# Surface-Bulk 2D Spin-Crossover Nanoparticles within Ising-like Model Solved by Using Entropic Sampling Technique

Catherine Cazelles [1,*], Mamadou Ndiaye [2,3], Pierre Dahoo [4], Jorge Linares [2,5,*] and Kamel Boukheddaden [2,*]

1   Université Paris-Saclay, UVSQ, IUT de Mantes en Yvelines, 78200 Mantes la Jolie, France
2   Université Paris-Saclay, UVSQ, CNRS, GEMAC, 78000 Versailles, France
3   Université Cheikh Anta Diop de Dakar, Fann, Dakar BP 5005, Senegal
4   Université Paris-Saclay, UVSQ, CNRS, LATMOS, 78290 Guyancourt, France
5   Departamento de Ciencias, Sección Física, Pontificia Universidad Católica del Perú, Av. Universitaria 1801, Lima 15088, Peru
*   Correspondence: catherine.cazelles@uvsq.fr (C.C.); jorge.linares@uvsq.fr (J.L.); kamel.boukheddaden@uvsq.fr (K.B.)

**Abstract:** We model the thermal effects in different 2D spin-crossover (SCO) square lattices within the frame of the Ising-like model using Monte Carlo entropic sampling (MCES) method to enhance the scan of macrostates beyond the most probable thermal ones. In fact, MCES allows access to the metastable states, and it is then well adapted to study thermal hysteresis properties. In this contribution, we distinguish, for the first time, the interaction between molecules located in bulk at the surface and those connecting the bulk and surface regions of an SCO lattice. In addition, an extra ligand field contribution is assigned to surface molecules through an interaction parameter L. In the absence of environmental effects on surface nanoparticles, a single thermal hysteresis loop increasing with the lattice size is simulated with a unique bulk and surface equilibrium temperature $T_{eq} = T_{eq}^{bulk} = T_{eq}^{surf}$. When environmental effects are accounted for, a two-step behavior associated with two hysteresis loops of widths $\Delta T_S$ (for the surface) and $\Delta T_B$ (for the bulk) with an intermediate plateau 14 K wide is obtained in the thermal dependence of the high-spin (HS) fraction for the $6 \times 6$ lattice. The surface and bulk equilibrium temperatures are then different, both decreasing towards lower values, and the L parameter controls the three states' behavior as well as the hysteresis loop interval. Size effects show that the equilibrium temperature is governed by the surface atoms for a small lattice size ($5 \times 5$) and by the bulk atoms for a large lattice size ($7 \times 7$). Moreover, a change in the size of the lattice results in a variation of the order–disorder (or Curie) temperature, $T_{O.D.}$, and the surface equilibrium temperature, $T_{eq}$, while only $T_{O.D.}$ changes in bulk.

**Keywords:** spin-crossover; nanoparticles; phase transition; Monte Carlo simulation; bulk-surface interactions

## 1. Introduction

SCO compounds [1–12] are among the most studied switchable molecular materials. They are the subject of worldwide research, both from theoretical and experimental approaches, with a focus on their potential technological applications in various domains, such as sensors, displays, switches, high-density reversible memories, solid-state coolants, nano-actuators, molecular spintronics, and various hybrid devices [13–18]. They provide the ability to control switching through various mechanisms and stimuli (such as light, thermal, pressure, external fields, etc.) [2,19–23] that trigger a transition between diamagnetic (low-spin state, LS, stable a low temperature) and paramagnetic states (high-spin state, HS, stable high-temperature). For d⁶ electronic configuration Fe(II) SCO complexes, the LS

state has a total spin S = 0, while the HS state has a spin S = 2. Because of the presence of occupied $e_g$ anti-bonding orbitals in the latter state, the bond strength between the Fe(II) and the ligand is weaker in HS-state than in LS-state. As a result, the Fe(II) metal center bond length is larger in the HS state by about 10% [1,2,19,21]. Moreover, as far as elastic properties are concerned, in the HS state, the SCO materials are softer and more malleable than in the LS state, where the lattice is more rigid due to its smaller volume.

The local volume change of each SCO unit along the spin transition, accompanying both elastic and electronic properties changes, leads to the bi-stable nature of this system. The cooperative hysteretic first-order thermal transition is due to the different elastic interactions, which can be highly anisotropic, mixing short-range (S-R) and long-range (L-R) effects combined with large electronic and vibrational entropy contributions of the HS-sate.

For a decade or so, trending synthesis and studies regarding SCO nanoparticles have aimed to unravel influences and effects which become prominent in their switching properties with size reduction.

To highlight the potential of nano SCO devices and materials, several studies are being developed to elucidate the effect of a particle's size, environment or surface on the ensemble of free nanoparticles, or nanoparticles embedded into polymeric matrices [24–26]. Thanks to these studies and experimental data, an underlying understanding of the manifestation of physical properties of the nanoparticles in accordance with their size, shape, and environment is developed, which will further help to propose models to mimic the experimental behaviors.

This work concerns modeling the thermodynamic effects of 2D SCO nanoparticles of several sizes accounting for specific interactions between the atoms on the surface, bulk, and interfaces [27–30]. The main purpose is to identify the key parameters playing a significant role in the control of the switching features of these complex materials to optimize the design of novel and specific systems with targeted physical properties.

The study of phase transitions through theoretical models at the microscopic level allows a better understanding not only of the physical mechanisms at the origin of the cooperativity in SCO compounds but also of the characteristics of the physical processes involved during the transformation of the materials, which typically incorporates various interconnected (electronic, mechanical, optical, etc.) properties. Indeed, previous experimental studies on SCO compounds have shown that macroscopic thermo-, photo- and piezo-transformations between LS and the HS states [21,22,31,32] are triggered by specific interactions at the microscopic level.

On the other hand, the molecular volume changes accompanying the LS to HS transitions delocalize over several unit cell parameters and interfere with each other. As a result, the SCO molecules may then interact through L-R effects caused by acoustic phonons and S-R couplings which can have an electronic or vibronic origin. While L-R interactions (which emanate from the elastic field contribution and mean-field interactions) stabilize homogeneous phases (HS or LS states) in most of the cases, they compete with nearest-neighbor short-range interactions which take place at the local scale and tend to stabilize inhomogeneous local structures. Consequently, a variety of thermal dependences of the HS fraction (Nhs) is observed: (i) first-order phase transitions with thermal hysteresis; (ii) gradual and stepwise one with two or three steps; (iii) incomplete spin transitions characterized by the presence of residual HS fractions at low-temperature; (iv) re-entrant transitions [19,22,32,33].

It is worth mentioning that the previous behaviors can also be obtained in SCO nanoparticles as the function of the size because of the effects induced by the competition between their bulk and surface contributions in addition to external stimuli (light, pressure, electric, etc.).

In the following, Section 2 is devoted to the model and principles of calculations. Section 3 describes the Monte Carlo Entropic Sampling method, while the results of our numerical simulations and the discussion are given in Section 4. Finally, in the last section, conclusions and future work are presented.

## 2. Ising like Model and Principles of Calculations

To model a 2D SCO square lattice of $N_x \times N_y$ size, one considers the Ising-like model. Wajnflasz and Pick (WP) [34] first proposed a two-state Ising-like model to describe SCO systems, where the high-spin (HS) and low-spin (LS) states of the molecule are associated with the eigenvalues $\sigma_i = +1$ and $\sigma_i = -1$, respectively, of a fictitious spin operator $\sigma$. The degeneracies of these two states are denoted by: $g_{HS}$ and $g_{LS}$, respectively. This WP model has been improved [35–37] to take into account short ($J$)–long ($G$) range interactions as well as the interaction ($L$) between the molecules at the surface and the surroundings (matrix effects).

The Hamiltonian is written as:

$$H = \frac{\Delta - k_B T \ln(g)}{2} \sum_{i=1}^{N} \sigma_i - G \sum_{i=1}^{N} \sigma_i \langle \sigma \rangle - J \sum_{\langle i,j \rangle} \sigma_i \, \sigma_j - L \sum_{k=1}^{M} \sigma_k \tag{1}$$

where $N$ is the total number of molecules, $g = g_{HS}/g_{LS}$, $\Delta$ the energy gap between the two states, and $\sigma_k$ is associated with the M molecules at the surface.

In this contribution, we distinguish, for the first time, the interactions between molecules of different regions: ($J_{BB}$) the interaction between the molecules at the bulk (blue-blue), ($J_{SS}$) between the molecules at the surface (red-red) and ($J_{BS}$) between the molecules at the bulk and those at the surface (red-blue), as depicted in Figure 1.

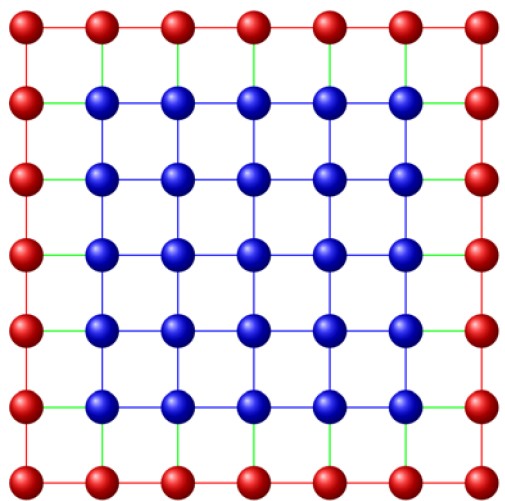

**Figure 1.** Schematic configuration of the interactions between atoms in the bulk (blue), surface (red), and bulk-surface (green).

The corresponding Hamiltonian writes:

$$H = \frac{\Delta - k_B T \ln(g)}{2} \sum_{i=1}^{N} \sigma_i - G \sum_{i=1}^{N} \sigma_i \langle \sigma \rangle - J_{bb} \sum_{bulk} \sigma_i \, \sigma_j$$

$$- J_{ss} \sum_{surf} \sigma_i \, \sigma_j - J_{bs} \sum_{B-S} \sigma_i \, \sigma_j - L \sum_{k=1}^{M} \sigma_k \tag{2}$$

$L$ is the result of the interaction between molecules at the surface with their environment, cast in its parametric form as:

$$H = -h \, m_t - J_{bb} \times s_b - J_{ss} \times s_s - J_{bs} \times s_{b-s} - L \, m_s \tag{3}$$

where, $h = \frac{\Delta - k_B T \ln(g) - 2\, G\, \frac{m_t}{N}}{2}$,

$$m_t = \sum_{i=1}^{N} \sigma_i \tag{4}$$

$$m_s = \sum_{k=1}^{M} \sigma_k \tag{5}$$

$s_b = \sum_{bulk} \sigma_i\, \sigma_j$ is the short-range correlation associated with the molecules in the bulk. Similarly, we define the short-range correlations function inside the surface sites and between surface and bulk sites as:

$s_s = \sum_{Surface} \sigma_i\, \sigma_j$ and $s_{bs} = \sum_{B-S} \sigma_i\, \sigma_j$, respectively.

The partition function Z and the thermal average value $\langle \sigma \rangle$ of $\sigma$, which is the average net fictitious magnetization, are calculated through the following expressions:

$$Z = \sum_{i=1}^{N_L} d\left(m_{t_i},\ s_{b_i},\ s_{s_i}, s_{bs_i},\ m_{s_i}\right) \exp(-\beta E_i) \tag{6}$$

where

$$E_i = -h m_{t_i} - J_{bb} s_{b_i} - J_{ss} s_{s_i} - J_{bs} s_{bs_i} - L m_{s_i} \tag{7}$$

and

$$\langle \sigma \rangle = \frac{\sum_{i=1}^{N_L} \frac{m_i}{N}\, d\left(m_{t_i},\ s_{b_i},\ s_{s_i}, s_{bs_i},\ m_{s_i}\right) \exp(-\beta E_i)}{Z} \tag{8}$$

where $N_L$ is the number of different configurations with the same five values $m_t$, $s_b$, $s_s$, $s_{bs}$ and $m_s$. The density of states $d(m_t,\ s_b,\ s_s, s_{bs}, m_s)$ is calculated by entropic sampling [38], and Equation (8) is solved by numerical techniques such as bisection. From the thermal average values $\langle \sigma \rangle$, the high-spin fraction ($Nhs$) is calculated by the relation:

$$Nhs = \frac{1 + \langle \sigma \rangle}{2}. \tag{9}$$

## 3. Monte Carlo Entropic Sampling

The Monte Carlo entropic sampling [38,39] is used to generate the table that contains the dimensionless macroscopic variables, $m_t$, $s_b$, $s_s$, $s_{bs}$, $m_s$ and their density, $d(m_t,\ s_b,\ s_s, s_{bs}, m_s)$. Therefore, we used the principles of MCES described by Shteto et al. [38], which consists of introducing into the detailed balance equation of the Monte Carlo (MC) procedure a suited biased distribution $P$ to favor configurations belonging to weakly degenerated macrostates and dampen those belonging to the highly degenerated macrostates.

The balance equation is given by:

$$P_i\, W(i \rightarrow j) = P_j\, W(j \rightarrow i), \tag{10}$$

While the biasing probability, chosen as the inverse of the desired restricted density of states, is given by:

$$P_i = \frac{1}{d\left(m_{t_i},\ s_{b_i},\ s_{s_i}, s_{bs_i},\ m_{s_i}\right)}. \tag{11}$$

Combining Equations (10) and (11), the balance equation becomes:

$$\frac{W(i \rightarrow j)}{W(j \rightarrow i)} = \frac{P_j}{P_i} = \frac{d\left(m_{t_i},\ s_{b_i},\ s_{s_i}, s_{bs_i},\ m_{s_i}\right)}{dd\left(m_{t_j},\ s_{b_j},\ s_{s_j}, s_{bs_j},\ m_{s_j}\right)}. \tag{12}$$

Since, in the first MC step, the density of the state $d\left(m_{t_i},\ s_{b_i},\ s_{s_i}, s_{bs_i},\ m_{s_i}\right)$ is unknown, we put all $d\left(m_{t_i},\ s_{b_i},\ s_{s_i}, s_{bs_i},\ m_{s_i}\right)$ equal to 1. Using $d\left(m_{t_i},\ s_{b_i},\ s_{s_i}, s_{bs_i},\ m_{s_i}\right)$ as a bias, an MC

sampling is run; it is termed a «Monte Carlo step» and yields a histogram of the frequency of the macrostates $H\left(m_{t_i}, s_{b_i}, s_{s_i}, s_{bs_i}, m_{s_i}\right)$, also written as $H_i\left(m_{t_i}, s_{b_i}, s_{s_i}, s_{bs_i}, m_{s_i}\right)$.

By construction, we have,

$$H_i(m_t, s_b, s_s, s_{bs}, m_s) \propto d_{i+1}(m_t, s_b, s_s, s_{bs}, m_s) \times (1/d_i(m_t, s_b, s_s, s_{bs}, m_s)). \quad (13)$$

The resulting restricted density of states is calculated after was applied the correction for the bias:

$$d_{i+1}(m_t, s_b, s_s, s_{bs}, m_s) \propto d_i(m_t, s_b, s_s, s_{bs}, m_s) \times H_i(m_t, s_b, s_s, s_{bs}, m_s) \quad (14)$$

The flat character of the histogram $H(m_t, s_b, s_s, s_{bs}, m_s)$ has a convenient convergence criterion. After the table that consists of $(m_t, s_b, s_s, s_{bs}, m_s)$ and $d(m_t, s_b, s_s, s_{bs}, m_s)$ is obtained, the exact partition function can be calculated using the expression (6).

## 4. Numerical Results and Analysis

In this study, we selected the thermodynamic parameters derived from experimental data of the prototype SCO complex [Fe(btr)$_2$(NCS)$_2$], btr = 4,4'-bis-1,2,4-triazole [12]. This particular material has the advantage of presenting a polymeric 2D structure made of weakly coupled SCO sheets, which is then well adapted to simulations with 2D square lattices. The enthalpy change ($\Delta H \approx 11$ kJ/mol) and the entropy change ($\Delta S \approx 50$ J/mol/K) lead respectively to the energy gap $\Delta = \Delta H/R \approx 1300$ K and to $\ln(g) = \Delta S/R \approx 6.01$, $R$ being the ideal gas constant. The equilibrium temperature of the system, $T_{eq}$, is deduced by $T_{eq} = \frac{\Delta/k_B}{(\ln(g))}$ which leads to $\approx 216.3$ K. To understand the behavior of the SCO compounds, let us also notice that $T_{up}$ and $T_{down}$ are, respectively, the ascending and the descending thermal transition temperatures. The average between $T_{up}$ and $T_{down}$ allows for knowing the equilibrium temperature of the system. This temperature is noted as $T_{eq}$ or $T_{1/2}$ because it corresponds to an HS fraction, $Nhs$, equal to 1/2. The difference $\Delta T = T_{up} - T_{down}$ represents the hysteresis width.

### 4.1. The Case L = 0

In this first case, the interactions between the molecules located on the surface and their environment are not considered, which amounts to setting $L = 0$ in the calculations. Hence, the effective ligand field is the same for bulk and surface atoms which leads to identical bulk and surface equilibrium temperatures, $T_{eq} = T_{eq}^{bulk} = T_{eq}^{surf} = \Delta/(k_B \ln(g)) = 216.3$ K.

As depicted in Figure 2, a single thermal hysteresis loop is observed for each particle size, and the width of the hysteresis loop $\Delta T = T_{up} - T_{down}$ increases as the size of the crystal lattice increases: $\Delta T \approx 19$ K in the case of the $5 \times 5$ system and $\Delta T \approx 26$ K in the case of the $7 \times 7$ system. This hysteretic behavior results from a competition between the order-disorder (or Curie) temperature designed by $T_{O.D.}$ and the equilibrium temperature $T_{eq}$, the fingerprint of a first-order transition. The increase of $\Delta T$ with size results from the fact that the order-disorder temperature of the pure Ising model, $T_{O.D.}$, increases with the size. By putting $\Delta/k_B = 0$, $L/k_B = 0$, $\ln(g) = 0$, $J_{bb}/k_B = 60$ K, $J_{ss}/k_B = 50$ K, $J_{bs}/k_B = 20$ K and $G = 172$ K, our calculations lead to the values gathered in Table 1.

**Table 1.** Evolution of the order-disorder temperature $T_{O.D.}$ and $\Delta T$ as a function of the size of the 2D system corresponding to Figure 1. The computational parameters are: $\Delta/k_B = 1300$ K, $L/k_B = 0$ K, $G/k_B = 172$ K, $J_{bb}/k_B = 60$ K, $J_{ss}/k_B = 50$ K, $J_{bs}/k_B = 20$ K and $\ln(g) = 6.01$. For $T_{O.D.}$, the values $\Delta/k_B = 0$ K and $\ln(g) = 0$ are used.

| Size of the System | $5 \times 5$ | $6 \times 6$ | $7 \times 7$ |
|---|---|---|---|
| $T_{O.D.}$ (K) | 296 | 308 | 316 |
| $\Delta T$ (K) | 19 | 22.6 | 26 |

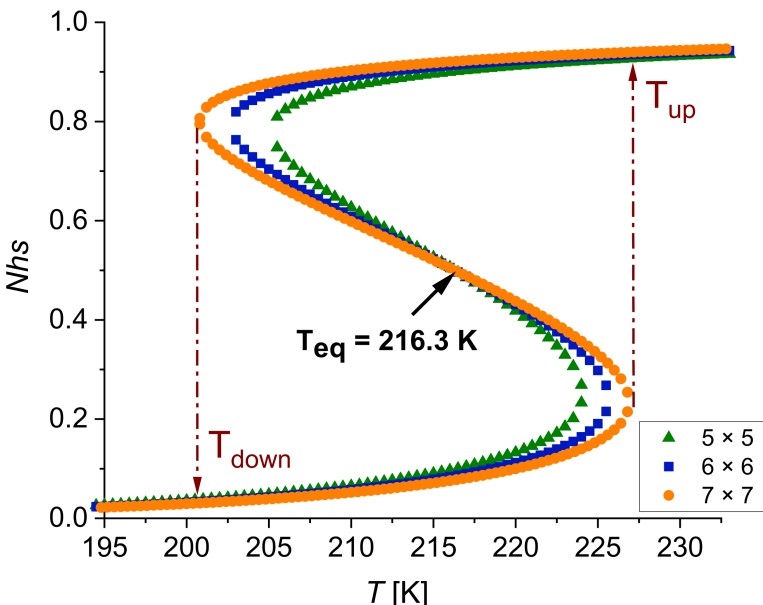

**Figure 2.** Thermal dependence of the HS fraction for square-shaped nanoparticles of different sizes: $5 \times 5$ (green up triangle), $6 \times 6$ (blue squares), $7 \times 7$ (orange circles). The computational parameters are: $\Delta/k_B = 1300$ K, $G/k_B = 172$ K, $J_{bb}/k_B = 60$ K, $J_{ss}/k_B = 50$ K, $J_{bs}/k_B = 20$ K, $L/k_B = 0$ K and $\ln(g) = 6.01$.

A first-order phase transition takes place when the condition $T_{O.D.} > T_{eq}$ is satisfied. This condition is fulfilled for systems $5 \times 5$, $6 \times 6$, and $7 \times 7$, as can be seen in Table 1, where $T_{O.D.}$ increases with size while $T_{eq}$ remains invariant. When the lattice size increases, the difference between $T_{O.D.}$ and $T_{eq}$ increases, and Figure 2 clearly shows that the width of the hysteresis loop increases simultaneously.

### 4.2. The Case $L \neq 0$

Next, the strength of the interactions between the surface molecules and the matrix is considered, and a positive value of $L$ has been used in the calculations. With a nonzero value of $L$, boundary effects increase, and, due to the matrix's contribution, the effective ligand field in the surface is weaker ($\Delta - 2L$) than in the bulk ($\Delta$). The surface $T_{eq}^{surf} = \frac{\Delta - 2L}{(k_B \ln(g))}$ and bulk $T_{eq}^{bulk} = \frac{\Delta}{(k_B \ln(g))}$ equilibrium temperatures become different.

In the case of a $6 \times 6$ lattice and with the parameters of Figure 3 ($G/k_B = 172$ K, $J_{bb}/k_B = 60$ K, $J_{ss}/k_B = 50$ K, $J_{bs}/k_B = 20$ K, $L/k_B = 290$ K), the simulations highlight a two-step behavior associated with two hysteresis loops, $\Delta T_S$ (for the surface) and $\Delta T_B$ (for the bulk). The first one $\Delta T_S$ extends in the range 148–154 K with values of $Nhs$ between 0 and 0.45, and the second one, $\Delta T_B$, in the range 175–183 K, with values of $Nhs$ between 0.62 and 1.0. An intermediate plateau 14 K wide appears for $Nhs$ between 0.5 and 0.62. It is associated with a mixture of HS and LS configurations and $T_{eq}^{surf}$ being less than $T_{eq}^{bulk}$, one can assert that molecules in the HS state are certainly those located on the surface. Indeed, the statistical weight of the surface for the $6 \times 6$ lattice is $\frac{5}{9} \approx 54\%$ ($N_{surf} = 20$, $N_{bulk} = 16$). The equilibrium temperature of the surface $T_{eq}^{surface}$ tends toward $\frac{\Delta - 2L}{(k_B \ln(g))} \approx 120$ K. The equilibrium temperature of the bulk $T_{eq}^{bulk}$, which should be equal to $\frac{\Delta}{(k_B \ln(g))} \approx 216.3$ K is shifted to lower temperatures on the one hand because of the interaction parameter $J_{bs}/k_B$ which connects the bulk and the surface and, on the other hand, because of the long-range interaction parameter $G/k_B$. A decrease in these two parameters would increase $T_{eq}^{bulk}$ and would decrease $T_{eq}^{surf}$ but would lead to the disappearance of this hysteretic behavior both in the bulk and on the surface.

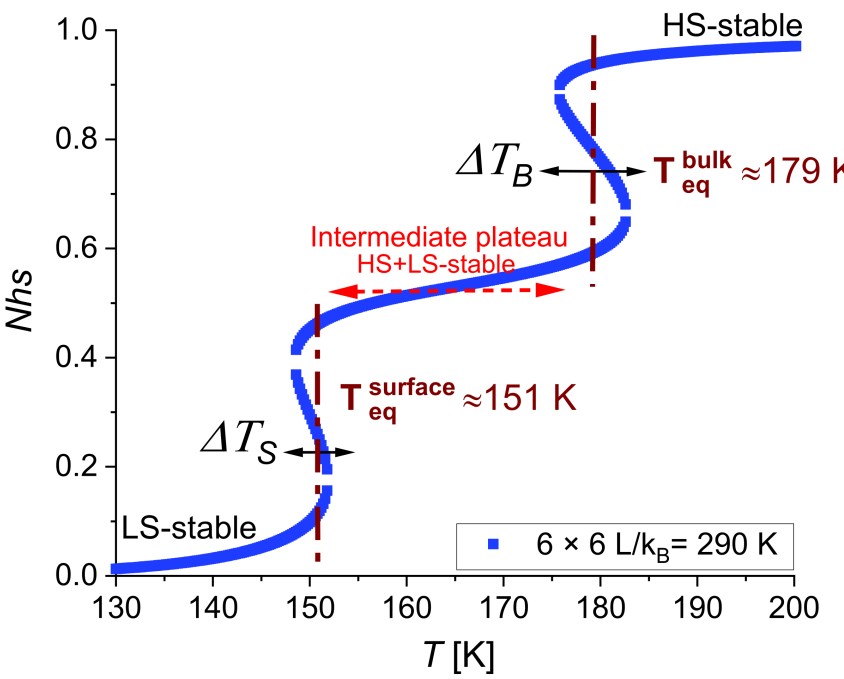

**Figure 3.** Thermal evolution of the HS fraction in an SCO system with size 6 × 6. The computational parameters are: $\Delta/k_B$ = 1300 K, $G/k_B$ = 172 K, $J_{bb}/k_B$ = 60 K, $J_{ss}/k_B$ = 50 K, $J_{bs}/k_B$ = 20 K, $L/k_B$ = 290 K and ln($g$) = 6.01.

The interplay between the two contributions, bulk and surface, is at the origin of this two-step transition. The thermal dependence of surface and bulk sites is represented separately in Figure 4.

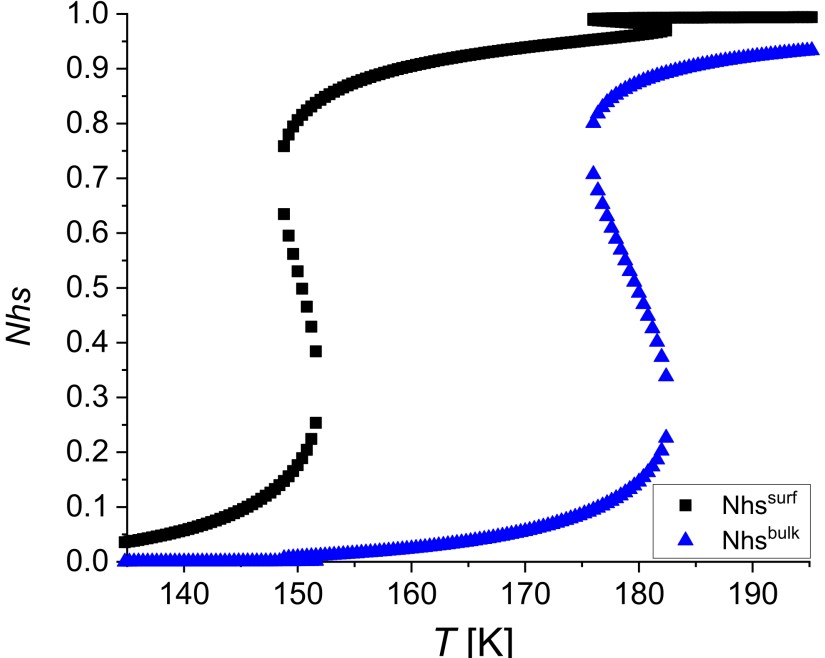

**Figure 4.** Thermal evolution of the HS fraction in an SCO system with size 6 × 6. Black squares: surface variation; Blue up-triangles: bulk variation. The calculation parameters are: $\Delta/k_B$ = 1300 K, $G/k_B$ = 172 K, $J_{bb}/k_B$ = 60 K, $J_{ss}/k_B$ = 50 K, $J_{bs}/k_B$ = 20 K, $L/k_B$ = 290 K, and ln($g$) = 6.01.

Figure 5 summarizes the thermal behavior of the HS fraction for the $6 \times 6$ lattice, where the value of the $L/k_B$ parameter has been gradually increased from 0 to 380 K. As already observed in Figure 2, the value $L/k_B = 0$ K leads to a single thermal hysteresis loop at the equilibrium temperature $T_{eq} = \Delta/(k_B \ln(g)) = 216.3$ K. For progressively higher values of the $L/k_B$ parameter, boundary effects increase, and a two-step transition emerges. The hysteresis loop for which *Nhs* is between 0 and 0.5 and corresponds to the surface molecules shifts towards low temperatures. The intermediate plateau, a mixture of HS and LS configurations, appears more clearly and gets wider ($\approx 60$ K when $L/k_B = 330$ K). This result shows that the $L$ interaction corresponding to the surface effect can control the three states' behavior as well as the hysteresis loop interval.

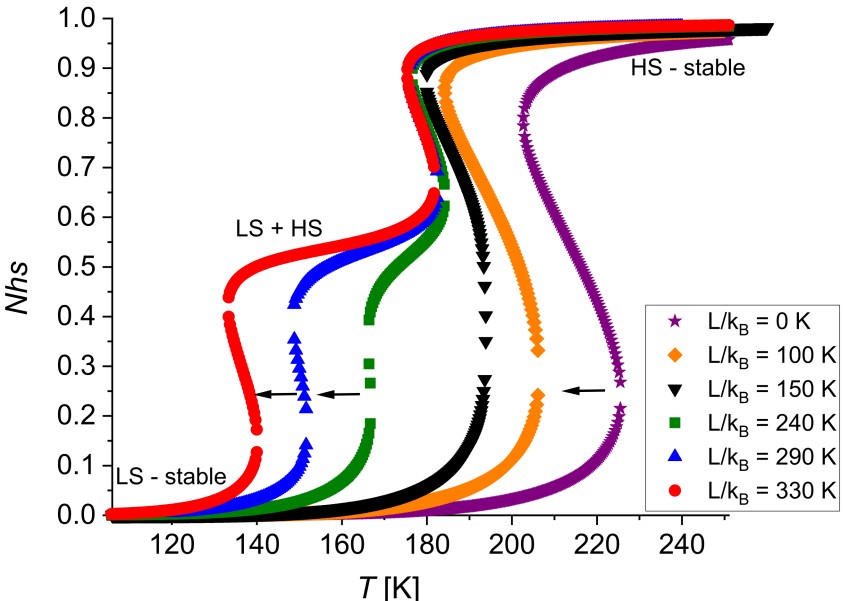

**Figure 5.** Simulated thermal evolution of the HS fraction *Nhs* for different values of the $L/k_B$ parameter in a $6 \times 6$ 2D SCO nanoparticle. Purple stars: $L/k_B = 0$ K; Orange diamonds: $L/k_B = 100$ K; Black down triangles: $L/k_B = 150$ K; Green squares: $L/k_B = 240$ K; Blue up triangles: $L/k_B = 290$ K; Red circles: $L/k_B = 330$ K. The parameters used in the calculations are: $\Delta/k_B = 1300$ K, $G/k_B = 172$ K, $J_{bb}/k_B = 60$ K, $J_{ss}/k_B = 50$ K, $J_{bs}/k_B = 20$ K and $\ln(g) = 6.01$.

The phase diagram of the $6 \times 6$ system is plotted in Figure 6, in the space coordinates, temperature versus $L/k_B$. It highlights the different types of transitions obtained in Figure 5 and reveals the existence of three domains of thermal behavior as a function of the surface parameter $L/k_B$. In region (i), corresponding to $L/k_B < 175$ K, the system presents a single hysteresis loop whose both equilibrium temperature is $T^1_{eq} \sim \frac{T^1_{up} + T^1_{down}}{2}$ and hysteresis width $\Delta T^1 = T^1_{up} - T^1_{down}$ are decreasing functions of $L/k_B$. The region (ii), with $175$ K $< L/k_B < 240$ K, depicts two-step transitions where one has a gradual nature (equilibrium temperature: $T^2_{eq}$) while the other belongs to the the hysteretic transition of region (i), for which the hysteresis width still decreases with the $L$ parameter. In region (iii), for which $L/k_B > 240$ K, the previous gradual transition becomes of first-order, and its associated equilibrium temperature, $T^2_{eq} \sim \frac{T^2_{up} + T^2_{down}}{2}$ decreases with L. Overall, the widths of the hysteresis loops $(T^1_{up} - T^1_{down})$ and $\left(T^2_{up} - T^2_{down}\right)$ correspond respectively to the bulk and surface contributions. When the $L$ value increases, the width of the hysteresis loops $\left(T^2_{up} - T^2_{down}\right)$ increases, whereas the width of the hysteresis loop $\left(T^1_{up} - T^1_{down}\right)$ remains unchanged in region (iii). Moreover, the equilibrium transition temperature $T^1_{eq} \sim \frac{T^1_{up} + T^1_{down}}{2}$ tends towards a constant value equal to $\sim 179$ K whereas $T^2_{eq} \sim \frac{T^2_{up} + T^2_{down}}{2}$ decreases, as was mentioned earlier.

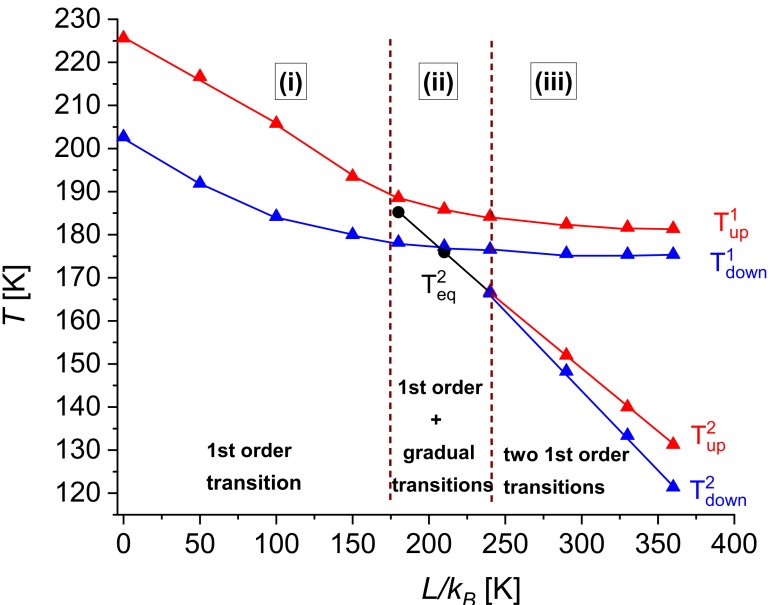

**Figure 6.** Phase diagram T versus ($L/k_B$) for a 2D 6 × 6 square-shaped system. The red and blue triangles correspond, respectively, to the upper and lower transitions for the heating ($T_{up}$) and cooling ($T_{down}$) temperatures of the thermal HS fraction. The black circles correspond to the equilibrium temperature ($T_{eq}$) of the gradual transition. The calculation parameters are: $\Delta/k_B = 1300$ K, $G/k_B = 172$ K, $J_{bb}/k_B = 60$ K, $J_{ss}/k_B = 50$ K, $J_{bs}/k_B = 20$ K and $\ln(g) = 6.01$.

For the value $L/k_B = 240$ K, and for the following set of interaction parameters, $J_{bb}/k_B = 60$ K, $J_{ss}/k_B = 60$ K and $J_{bs}/k_B = 50$ K, two superimposed hysteresis loops combining bulk and surface responses leading to the following processes (LS) ↔ (LS) + (HS) and (LS) + (HS) ↔ (HS) can be observed on heating, while on cooling, one has a one-step transition from (HS) to (LS). This leads to a non-symmetric thermal hysteresis: a two-step transition on the heating mode and a one-step on the cooling mode, as shown in Figure 7b. Another result shown in Figure 7a, concerns the two hysteresis overlap in the temperature range $T'' - T' \approx 3$ K and leads to the coexistence of three stable states, which are the result of a competition between the environmental effect and the internal "elastic" interactions.

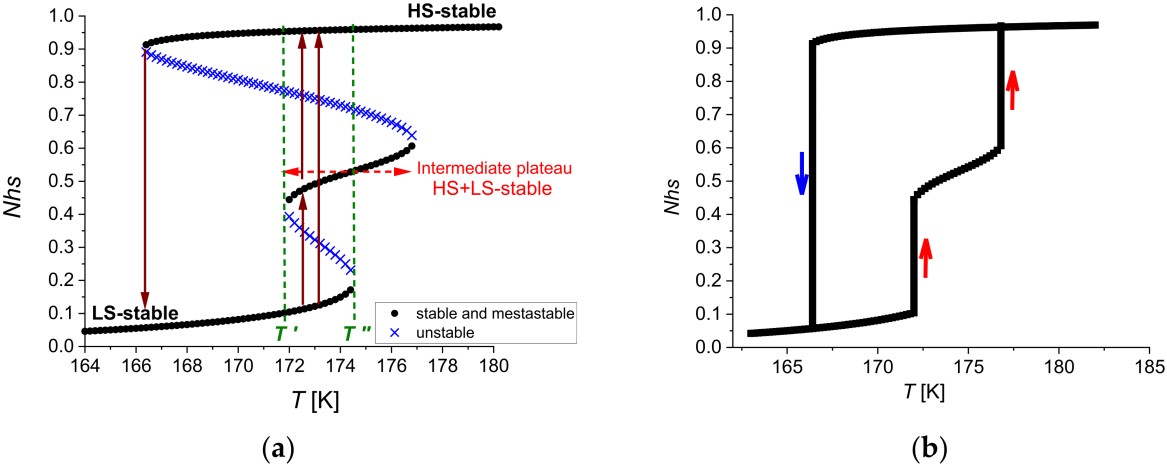

**Figure 7.** Thermal evolution of the HS molar fraction in a square SCO lattice with size 6 × 6. The computational parameters are $L/k_B = 240$ K, $G/k_B = 172$ K, $J_{bb}/k_B = 60$ K, $J_{ss}/k_B = 60$ K, $J_{bs}/k_B = 50$ K and $\ln(g) = 6.01$. (**a**) All the solutions are shown. The curve indicates the three states between $T'$ and $T''$. (**b**) A two-step transition on the heating mode and one-step on the cooling mode is shown.

### 4.3. Size Effects under Temperature

In this part, the evolution of the HS fraction, as a function of temperature, is studied for different sizes of 2D square lattice ($5 \times 5$, $6 \times 6$, and $7 \times 7$). The ratio $r$ between the surface atoms $N_{surf}$ and the total number of atoms $N$ is defined, and the values of $r$ are gathered in Table 2. This ratio $r$ is significant to highlight the influence of the environment, correlated to the $L/k_B$ interaction term. In fact, upon reducing the lattice size, $r$ increases, the $L/k_B$ term acts on a larger number of atoms, and, therefore, the surface atoms control the thermal behavior of the compound. To fully understand the thermal behavior of the present system, it is worth mentioning that the difference ($T_{O.D.} - T_{eq}$) controls the nature of the transition as well as the width and position of the thermal hysteresis associated with a first-order phase transition taking place when the condition $T_{eq} < T_{O.D.}$ is fulfilled.

**Table 2.** Values of the ratio $r$ between the number of surface atoms $N_{surf}$ and the total number of atoms $N$ for $5 \times 5$, $6 \times 6$, and $7 \times 7$ lattices.

| System's Size | $N$ | $N_{bulk}$ | $N_{surf}$ | $r = N_{surf}/N$ |
|---|---|---|---|---|
| $5 \times 5$ | 25 | 9 | 16 | 0.64 |
| $6 \times 6$ | 36 | 16 | 20 | 0.55 |
| $7 \times 7$ | 49 | 25 | 24 | 0.48 |

Let us specify that a change in the size of the lattice results in a variation of $T_{O.D.}$ and $T_{eq}^{surf}$ whereas the equilibrium temperature of the core $T_{eq}^{bulk} = \frac{\Delta}{k_B \ln(g)} \approx 216.3$ K is constant. A decrease in the lattice size shifts $T_{O.D.}$, $T_{eq}^{surf}$ and the equilibrium temperature $T_{eq}^{syst}$ (defined for $Nhs = \frac{1}{2}$) of the system towards lower temperatures. Moreover, $T_{eq}^{surf}$ and $T_{eq}^{syst}$ decrease faster than $T_{O.D.}$.

The curves obtained for the different lattice sizes are shown in Figure 8, and the widths of the thermal hysteresis $\Delta T_s = T_{up}^{surf} - T_{down}^{surf}$ and $\Delta T_b = T_{up}^{bulk} - T_{down}^{bulk}$ are gathered in Table 3 for the $5 \times 5$, $6 \times 6$ and $7 \times 7$ systems.

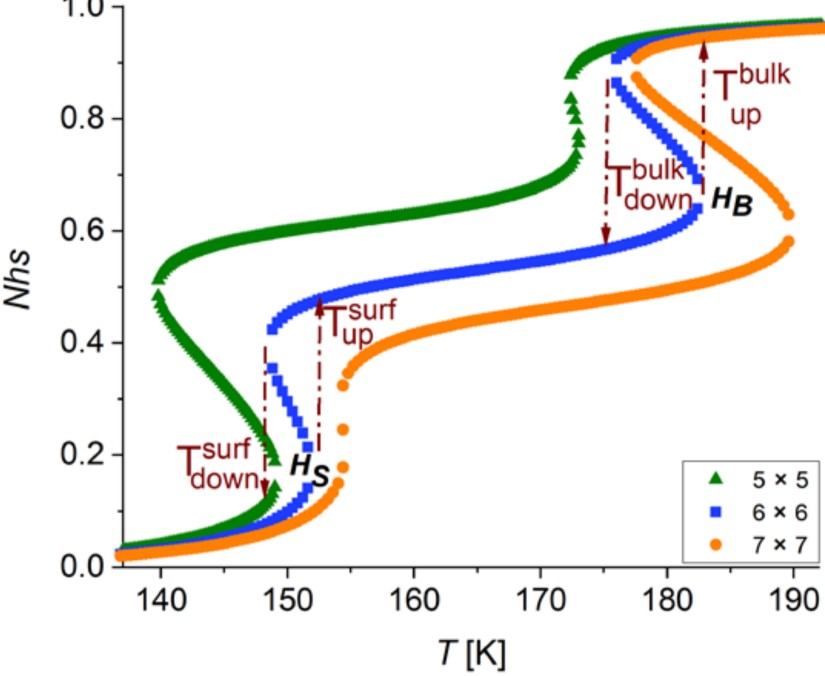

**Figure 8.** Thermal evolution of the HS fraction for different lattice sizes: $5 \times 5$ (green up triangle), $6 \times 6$ (blue squares), $7 \times 7$ (orange circles). The computational parameters are: $\Delta/k_B = 1300$ K, $L/k_B = 290$ K, $G/k_B = 172$ K, $J_{bb}/k_B = 60$ K, $J_{ss}/k_B = 50$ K, $J_{bs}/k_B = 20$ K and $\ln(g) = 6.01$.

**Table 3.** Evolution of the width of the thermal hysteresis $\Delta T = T_{up} - T_{down}$ as a function of the size of the 2D system corresponding to Figure 3. The computational parameters are: $\Delta/k_B$ = 1300 K, $L/k_B$ = 290 K, $G/k_B$ = 172 K, $J_{bb}/k_B$ = 60 K, $J_{ss}/k_B$ = 50 K, $J_{bs}/k_B$ = 20 K and $\ln(g)$ = 6.01.

| System's Size | $\Delta T_s = T_{up}^{surf} - T_{down}^{surf}$ (K) | $\Delta T_b = T_{up}^{bulk} - T_{down}^{bulk}$ (K) |
|---|---|---|
| 5 × 5 | 10.0 | 1.0 |
| 6 × 6 | 4.4 | 7.3 |
| 7 × 7 | 0.0 | 12.5 |

For a small lattice size (5 × 5) corresponding to a large value of *r* (see Table 2) and owing to the $L/k_B$ term, the equilibrium temperature of the system is governed by the surface atoms. The significant gap between $T_{eq}^{surf}$ and $T_{O.D.}$ and the condition $T_{eq}^{surf} < T_{O.D.}$ leads to a large hysteresis width $\Delta T_s$ (see Table 3). In the bulk, the gap between $T_{eq}^{bulk}$ and $T_{O.D.}$ is smaller, which leads to a less significant hysteresis phenomenon $\Delta T_b$ (see Table 3).

For a larger lattice size (7 × 7) corresponding to a smaller value of *r*, the equilibrium temperature of the system is mainly governed by the bulk atoms. On the surface, the hysteresis phenomenon disappears, and an abrupt transition is observed. In the bulk, on the contrary, due to the large number of interactions, a significant gap between $T_{eq}^{bulk}$ and $T_{O.D.}$ and the condition $T_{eq}^{bulk} < T_{O.D.}$ leads to a large hysteresis $\Delta T_b$. The surface and the bulk, therefore, have opposite behaviors when the size of the system varies.

*4.4. The Case $J_{bs} = 0$*

To get further in the study of the thermal properties of our 2D bulk-surface system, the case $J_{bs} = 0$ was investigated for several system sizes, and the results are reported in Figure 9 for a 6 × 6 SCO compound. When the short-range interactions $J_{bs}$ between the bulk and the surface are not considered in the simulations; only the long-range interactions *G* connect these two entities. The simulations then show that the $H_s$ hysteresis (shell) is shifted toward lower temperatures, whereas the $H_B$ hysteresis (bulk) is shifted toward higher temperatures. Moreover, the intermediate plateau, constructed by a sequence of +1 and −1 configurations, is wider and extends between $T \approx 148$ K and $T \approx 183$ K. These results are also observed in the case of the 5 × 5 and 6 × 6 lattices.

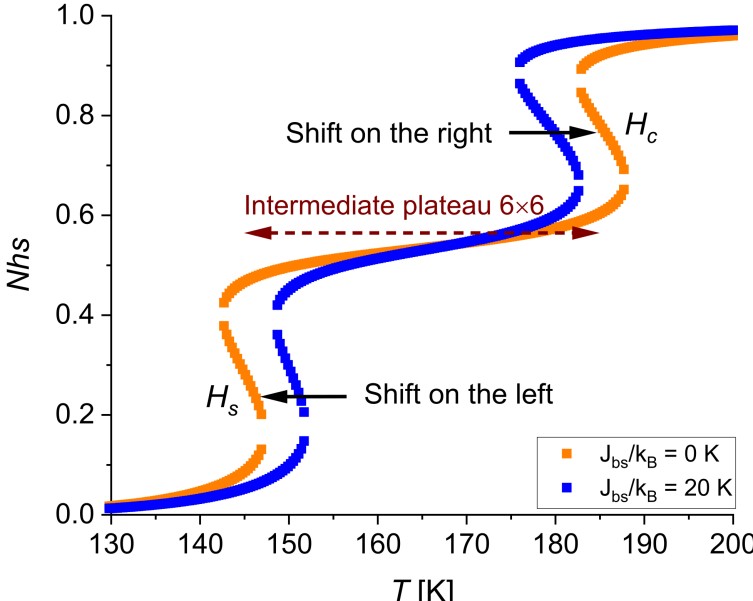

**Figure 9.** Thermal evolution of the HS fraction in the case of a 6 × 6 lattice for $J_{bs}/k_B$ = 20 K (blue squares) and $J_{bs}/k_B$ = 0 K (orange squares). The computational parameters are: $\Delta/k_B$ = 1300 K, $L/k_B$ = 290 K, $G/k_B$ = 172 K, $J_{bb}/k_B$ = 60 K, $J_{ss}/k_B$ = 50 K and $\ln(g)$ = 6.01.

## 5. Conclusions

In summary, we presented an adapted two-state Ising-like Hamiltonian allowing us to model SCO nanoparticles by considering L-R and S-R interactions as well as specific surface effects. In particular, three types of S-R interactions are considered: between molecules in the bulk, on the surface, and in the surface-bulk region. The model is solved within the frame of the MCES technique, which provides access to the density of macro-states for different sizes of nanoparticles and interaction parameter values. When the interactions between surface molecules and their surrounding environment are considered to be negligible ($T_{eq} = T_{eq}^{bulk} = T_{eq}^{shell}$), the system stabilizes first-order single thermal hysteresis loops for different particle sizes, with the widths of the loops increasing the function of the nanoparticle size. The appearance of a first-order transition is due to the increase of the order-disorder (or Curie) temperature of the pure Ising model ($T_{O.D.}$) as a function of the size of the compound compared to the transition temperature $T_{eq}$, on one hand, and to the competition between short- and long-range interactions on the other hand. Indeed, when the size of the nanoparticles increases, the long-range intermolecular interactions which stabilize the HS or LS state competes with the short-range interactions whose strength depends on the size of the lattice and monitors the width of the thermal hysteresis.

Then, considering the surrounding environment contribution, the effective ligand field on the surface is weaker than in the bulk. This leads to a lower equilibrium temperature at the surface than that obtained in the bulk. This difference in equilibrium temperatures naturally stabilizes a two-step transition with two hysteretic first-orders, one for the surface ($H_S$) and the other for the bulk ($H_B$). When the equilibrium temperature of the surface is lower than that of the bulk, the thermal transition of the bulk on cooling takes place before that of the surface, leading to a stabilization of an intermediate plateau of 14 K wide between *Nhs* 0.5 and 0.62, which is associated with a mixture of HS shell and LS core configurations. Compared to the case $L/k_B = 0$ (no surface effect), the decrease of the equilibrium temperature of the bulk $T_{eq}^{bulk}$ to lower temperatures (in the case of Bulk-Surface configurations) is due to the interaction parameter $J_{bs}$ which connects the bulk and the surface and the long-range interaction parameter $G$. When the value of the $L$ parameter is gradually increased from 0 to 380 K, the hysteresis loop and equilibrium temperature corresponding to the shell component shift towards lower temperatures. This results in the increase of the width of the intermediate plateau ($\Delta T \approx 60$ K), which becomes flatter and flatter. The thermal dependence of the HS fraction is controlled by $L$ parameter. Increasing the interaction parameters of the bulk ($J_C$), surface ($J_S$) and bulk-surface ($J_{CS}$), leads to a coexistence of three "stable" states (multi-step transition) due to the competition between the environmental effect and the internal "elastic" interactions. The effect of the lattice size is also analyzed. For small particle sizes corresponding to a large value of the ratio, $r$, of atoms in the shell and the total number of atoms, the hysteresis loop, $H_s$, for the surface increases, while the hysteresis loop, $H_c$, for the bulk decreases due to the predominance of the interactions on the surface, which control the thermal behavior of the compound. For a relatively larger particle lattice size (small value of $r$), the hysteresis in the surface vanishes, and the transition becomes abrupt, while that of the bulk becomes larger due to the increase of the interactions in the bulk, which control the thermal behavior of the compound. Indeed, when the size of the system varies, the surface and the bulk have opposite behaviors, which establishes a direct competition between these two nanoparticle components. In addition, when the short-range interactions between the bulk and the surface $J_{bs}$ are not considered in the simulations; only the long-range interactions $G$ connect these two entities. In this case, the results show that the surface hysteresis is shifted downward, whereas the bulk hysteresis is shifted upward, inducing a widened intermediate plateau. The present work will be extended to other lattice shapes (rectangular, hexagonal, and cubic) and other symmetries (triangular, square, hexagonal), leading to different atomic coordination numbers.

**Author Contributions:** Conceptualization, J.L. and K.B.; Methodology, J.L. and K.B.; Software, C.C. and J.L.; Validation, C.C., M.N., P.D., J.L. and K.B.; Formal analysis, M.N., P.D., C.C., J.L. and K.B.; Investigation, J.L. and K.B.; Writing—original draft, C.C., M.N., P.D., J.L. and K.B.; Supervision, J.L. and K.B. All authors have read and agreed to the published version of the manuscript.

**Funding:** This research received no external funding.

**Institutional Review Board Statement:** Not applicable.

**Informed Consent Statement:** Not applicable.

**Data Availability Statement:** Not applicable.

**Acknowledgments:** This research was funded by the CNRS and the Université de Versailles St-Quentin member of the Université Paris-Saclay, the ANR project Mol-CoSM no. ANR-20-CE07-0028-02. We thank all of them for their financial support.

**Conflicts of Interest:** The authors declare no conflict of interest.

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
