# Peer review of "Surface-Bulk 2D Spin-Crossover Nanoparticles within Ising-like Model Solved by Using Entropic Sampling Technique"

_magnetochemistry, doi:10.3390/magnetochemistry9030061_

Round 1

Reviewer 1 Report

I have read the script as a practicising experimentalist in the topic: In the submitted manuscript, the authors modeled SCO in 2D nanaoparticles by considering several scenarios. The authors elucidate how the interactions between the surface and core parts of the nanoparticle and the interaction between the surface and environment defines the nature of the spin-state switching. Scenariors leading to one-step to several-step transitions have been discussed. The group is well known for the work of the kind presented in the script, and the submitted study is competently done and well-written. The following are a few minor remarks, which need to be adressed in the revised version.

Please define TO.D. in the abstract.

Spin crossover or spin-crossover?

Please check line 393: "This leads to a lower the transition temperature"

The SCO phenomenon and the possibility of applications in several fields have been proposed. Despite morethan 75 years of research, no realistic applications based on SCO complexes have been demonstrtaed. In this light, the introduction could be toned down a bit. Knowledge creation is the most important aspect of basic science, and the study rather nicely elucidates the factors contributing to the observation of stepwise SCO.

Can one extrapolate the results to undestand stepwise SCO in larger molecular crystals?

What are those environmental factors that interact with the switching unints at the surface?

Why only square lattices are considered?

Author Response

Dear Reveiwer,

We thank you for all your comments which have helped us to improve our new version.

Reviewer 2 Report

The scientific content (theoretical in nature) of this ms is very interesting and this work thus deserves-according to my opinion-acceptance and publication in MAGNETOCHEMISTRY. I am sure that the paper will attract the interest of scientists working in the general area of the Spin Crossover (SCO) phenomenon, and especially in aspects of this phenomenon at the nanoscale. Also, I do believe that the article will receive a respectable number of citations in the future. Salient features of this work-which justify my proposal for acceptance-are: (a) The thermal effects in different SCO square lattices have been modeled within the Ising-like approach using the method of Monte Carlo entropic sampling (for different nanoparticle sizes and for different values of the interaction parameter). (b) The authors were able to distinguish-for the first time-the interactions between molecules in the bulk, at the surface and those linking the surface and the bulk; and (c) An additional ligand field contribution has been assigned to surface molecules through an interaction parameter. The ms is well written and nicely organized. The quality of figures and equations is high, and the references list covers the topic under study more than satisfactorily.

Based on the above mentioned, I am glad because I can propose acceptance of this fine piece of research in MAGNETOCHEMISTRY. I do not have scientific points to raise.  Minor revision points/comments/suggestions to be taken into account by the authors:

(1)    Section 5, “Conclusions”: The perspectives and broad future scope of this work should be outlined.

(2)    Section 4: A schematic drawing of the structure of [Fe(btr)2(NCS)2] should be presented; such a scheme will be helpful for the non-familiar readers.

(3)    The authors should mention the criteria behind the choice of this particular SCO iron(II) complex; and

(4)    Equations (13) and (14) should be better arranged.

Author Response

We thank you for all your comments which have helped us to improve our new version.

Reviewer 3 Report

The authors present an article about Surface-Bulk 2D Spin Crossover Nanoparticles within Ising-Like Model Solved by Using Entropic Sampling Technique.

Spin crossover molecular materials (SCO) have been the subject of particular attention for several decades because of their interesting HS-LS bistability properties in many applications. Each year, many articles deal with this subject from an experimental point of view or in numerical simulations, which is the case of this paper, with always new points addressed as this field of research is rich. The authors prove it once again by proposing an original study and a new contribution to the existing literature: thermal effects in different 2D SCO square lattices are modeled using Monte Carlo entropic sampling method with a specificity : the short-range interactions between molecules located in the bulk, at the surface and those connecting the bulk and surface regions are taken into account.

The authors present the subject in the introduction which makes it possible to understand the context of the study and the contributions of this new article in the field of SCO materials.

Model and principles of calculations are detailed in the next sections 2 and 3.

Section 4 presents numerical results and analysis. They are well explained and detailed. Figures help to follow the different simulations made in specific cases of interaction between molecules and of lattice sizes highlighting the role of the surface or the bulk regions in the evolution of the thermal transition temperature. Readers who know the field will immediately notice the accuracy of the model.

The authors summarize the main results in the conclusion by highlighting the different scenarios analyzed.

I think this article can be considered for publication in the Journal Magnetochemistry.

Author Response

(The authors gave the same response as above.)
